# Interplay between Intrinsic and Innate Immunity during HIV Infection

**DOI:** 10.3390/cells8080922

**Published:** 2019-08-17

**Authors:** Louis Bergantz, Frédéric Subra, Eric Deprez, Olivier Delelis, Clémence Richetta

**Affiliations:** Laboratoire de Biologie et Pharmacologie Appliquée, Ecole Normale Supérieure Paris-Saclay, CNRS UMR8113, IDA FR3242, Université Paris-Saclay, F-94235 Cachan, France

**Keywords:** HIV-1, host restriction factors, innate immune sensing, immune responses, interferon, viral counteraction

## Abstract

Restriction factors are antiviral components of intrinsic immunity which constitute a first line of defense by blocking different steps of the human immunodeficiency virus (HIV) replication cycle. In immune cells, HIV infection is also sensed by several pattern recognition receptors (PRRs), leading to type I interferon (IFN-I) and inflammatory cytokines production that upregulate antiviral interferon-stimulated genes (ISGs). Several studies suggest a link between these two types of immunity. Indeed, restriction factors, that are generally interferon-inducible, are able to modulate immune responses. This review highlights recent knowledge of the interplay between restriction factors and immunity inducing antiviral defenses. Counteraction of this intrinsic and innate immunity by HIV viral proteins will also be discussed.

## 1. Introduction

To counteract a viral infection, immune as well as non-immune cells have a wide variety of cellular factors able to fight the virus. In the case of human immunodeficiency virus 1 (HIV-1) infection, multiple cellular proteins called restriction factors constitute a first line of antiviral defense by blocking specific steps of the virus cycle including entry, uncoating, reverse transcription and budding [1]. More generally, restriction factors are proteins that directly inhibit the replication of a variety of viral families by targeting conserved mechanisms shared by viruses. Restriction factors against HIV-1 include the well-known APOBEC3G (apolipoprotein B mRNA-editing enzyme catalytic subunit-like 3G), SAMHD1 (sterile alpha motif and histidine aspartate domain containing protein 1), Tetherin/BST-2 (bone marrow stromal antigen 2), tripartite motif (TRIM)5α [2,3,4,5,6,7,8] and proteins that have been characterized more recently, such as IFITM (interferon-inducible transmembrane proteins), Mx2 (Myxovirus resistance protein 2), SERINC3/5 (serine incorporator 3/5), Schlafen 11 or ISG15 (interferon-stimulated gene 15) for instance [9,10,11,12]. Viral restriction exerted by these host factors has the property to be immediate and direct and is defined as “intrinsic antiviral immunity” to distinguish it from the action of antiviral effectors that need to be induced by innate immunity. Indeed, intrinsic immune viral restriction factors are usually preexisting in certain cell types rendering a cell non-permissive to a specific class or species of viruses.

In addition, viral infections are also sensed by receptors of the innate immune system called pattern recognition receptors (PRRs). PRRs include several types of receptors such as Toll-like-receptors (TLR) which are transmembrane receptors expressed on the cell surface or in endosomal compartments, cytosolic receptors of the retinoic acid-inducible gene I (RIG-I)-like receptor (RLR) family and the nucleotide-binding oligomerization domain (NOD)-like receptors (NLR). PRRs recognize conserved motifs expressed on pathogens called pathogen-associated-molecular patterns (PAMPs). Detection of PAMPs by PRRs induces signaling cascades and leads to the transcription of pro-inflammatory cytokines and type I interferons (IFN-I), the most powerful antiviral effectors. Once produced by cells in which PRRs have been activated, IFN-I are secreted and bind, in an autocrine or paracrine manner, to the membrane receptor IFNAR (IFN-α/β receptor chain) to induce signal transduction driving the transcription of interferon-stimulated genes (ISGs). There are hundreds of ISGs encoding antiviral proteins such as the protein kinase R (PKR) or the 2′-5′-oligoadenylate synthetase 1 (OAS1) [13,14]. Thus, ISGs are crucial to switch cells into an antiviral state. Interestingly, even if HIV restriction factors have been described as intrinsic immune effectors, most of them are IFN-inducible and can be defined as ISGs linking intrinsic and innate immunity.

Moreover, by exerting their antiviral functions or by additional pathways, some restriction factors have been described to shape innate and adaptive immune responses highlighting an intimate relationship between restriction factors and immunity [15,16]. In this review, we summarize the recent knowledge of the interplay between restriction factors and immunity focusing only on restriction factors that belong to the ISG family. After a description of restriction factors antiviral mechanisms and their induction by IFN, we will discuss how they can modulate immunity, in particular, innate immune responses. Moreover, since HIV-1 proteins are able to counteract many restriction factors, we will also mention viral mechanisms used to escape from intrinsic and innate immunity. Taken together, this review will allow a better understanding of the intricate interplay between restriction factors-mediated intrinsic immunity and innate immunity.

## 2. Activation of Innate Immunity Following HIV Sensing

HIV-1 infection is sensed by the immune system thanks to the recognition of several HIV PAMPs by PRRs. These PAMPs are envelope glycoproteins, the capsid, viral single-stranded RNA (ssRNA) genome and nucleic acids produced during reverse transcription. Several classes of PRRs are involved in the recognition of HIV PAMPs: TLR, RIG-I, but also other cytosolic receptors such as cGAS (cyclic guanosine monophosphate-adenosine monophosphate synthase)-STING (stimulator of interferon genes) and IFI16 (interferon gamma inducible protein 16)-STING. Moreover, HIV-1 is recognized by NLR activating the inflammasome.

Before giving details about HIV-1 sensing by PRRs, we will summarize the molecular pathways leading to cytokines and IFN-I production after PAMP recognition.

### 2.1. General Overview of Pattern Recognition Receptors Signaling after Pathogen Associated Molecular Patterns Recognition

Due to their localization on the cell surface and in endosomes, TLRs recognize both extracellular pathogens and nucleic acids derived from viruses. TLRs share a conserved structure comprising an ectodomain, composed of a leucine-rich repeat domain (LRR), a transmembrane segment and a cytoplasmic Toll/Interleukin-1 receptor domain (TIR). Upon interaction of PAMPs with their LRR domain, TLRs undergo homo- or hetero-dimerization inducing the recruitment and the binding of cytoplasmic adaptor molecules to their TIR domains. The adaptors used by human TLRs to initiate the signal transduction cascade are the myeloid differentiation factor 88 (MyD88), the TIR domain-containing adaptor-inducing IFN-β (TRIF), the MyD88-adaptor-like (MAL, known also as TIRAP) and the TRIF-related adaptor molecule (TRAM; known also as TICAM2) [17]. MyD88 and TRIF mediate the signaling cascade by recruiting downstream kinases whereas MAL/TIRAP and TRAM translocate MyD88 or TRIF on activated TLRs. Depending on the adaptor used, two intracellular pathways are engaged. The MyD88-dependent pathway is used by all TLRs except TLR3 which signals through the TRIF-dependent pathway. TLR4 can both recruit MyD88 or TRIF and is able to induce IFN-I by using the two signaling pathways (Figure 1).

The MyD88-dependent pathway leads on one hand, to the activation of the transcription factor NF-κB (nuclear factor-kappa B) involved in the transcription of pro-inflammatory cytokines such as TNF-α (tumor necrosis factor-α), interleukin (IL)-1β and IL-6 and, on the other hand, to the production of IFN-I (Figure 1). After its binding to the TIR domain of activated TLRs, MyD88 recruits and activates the serine-threonine protein kinases IRAK (interleukin-1 receptor-associated kinase), which then recruit the E3 ubiquitin ligase TRAF6 (TNF receptor associated factor 6). The E2 ligase TRICA1 in cooperation with TRAF6 generates a scaffold of polyubiquitin chains on TRAF6 itself and on the protein NEMO (IKKγ) leading to activation of the serine-threonine kinase TAK1 (transforming growth factor beta-activated kinase 1). TAK1 then acts in two signaling pathways. First, TAK1 activates mitogen-activated protein kinases (MAPKs) leading to activation of AP-1 family transcription factors. Secondly, TAK1 is able to induce the NF-κB pathway. NF-κB is a nuclear factor that is sequestered into the cytoplasm by the inhibitory factor IκB (Figure 1). TAK1 phosphorylates and activates the IκB kinase (IKK) complex composed of the three proteins IKKα, IKKβ and IKKγ (NEMO). In return, IKK phosphorylates the inhibitory factor IκB driving its proteasomal degradation and inducing the release of NF-κB and its subsequent translocation into the nucleus where it activates the transcription of pro-inflammatory cytokines [18,19]. Induction of IFN-I through MyD88 is dependent on the activation of interferon-regulatory transcription factors (IRF). In the absence of activation, IRFs are retained in an inactive form into the cytoplasm and need to be phosphorylated on serine and threonine residues on their carboxy termini to become active. The most important IRFs involved in TLR signaling are IRF3 and IRF7. MyD88 can directly interact and activate IRF7 which translocates into the nucleus to activate transcription of IFN-α and IFN-β [19].

For TLR3 which does not use the MyD88-dependent pathway, but the adaptor protein TRIF, TRIF binds and activates the kinases IKKε and TBK1 (tank-binding kinase 1) which in turn, phosphorylate IRF3 to induce the production of IFN-β (Figure 1).

In addition to TLRs, cytoplasmic receptors, and in particular RLRs, drive the production of IFN-I after the recognition of PAMPs from intracellular pathogens. RLRs comprise two members: the retinoic acid-inducible gene I (RIG-I) and the melanoma differentiation-associated protein 5 (MDA5). After the recognition of nucleic acids (dsRNA or ssRNA) by RIG-I or MDA-5, their CARDs (caspase activation and recruitment domains) interact with the CARD-containing adaptor mitochondrial antiviral-signaling (MAVS) protein localized in the outer mitochondrial membrane (Figure 1). MAVS then activates the IKK-related kinases, TBK1/IKKi, leading to activation of IRF3/IRF7 and the production of IFN-α and IFN-β. MAVS also activates NF-κB and the production of pro-inflammatory cytokines through the recruitment of the protein FADD (Fas-associated protein with death domain.) [19].

The production of IFN-I is crucial for the establishment of an antiviral state since it leads to the transcription of antiviral effectors in cells activated by PRRs but also in neighboring cells. IFN-I are encoded by numerous genes clustered on chromosome 9 and include IFN-α, IFN-β, IFN-ε, IFN-κ and IFN-ω. Once produced by cells in which PRRs have been activated, IFN-I are secreted and induce paracrine or autocrine signaling by binding to the membrane IFN-α/β receptor (IFNAR) composed of two subunits: IFNAR1 and IFNAR2. Then IFNAR phosphorylates and activates the Janus kinase 1 (JAK1) and the non-receptor tyrosine kinase 2 (TYK2), which mediate the phosphorylation and the dimerization of the transcription factors STAT1 (signal transducer and activator of transcription) and STAT2 [20]. Dimerized STAT1 and STAT2 then recruit IFN-regulatory factor 9 (IRF9) to form STAT1-STAT2-IRF9 tri-complex called ISGF3. After its translocation into the nucleus, ISGF3 binds to a conserved sequence named IFN-stimulated response elements (ISRE) and activates the transcription of IFN-stimulated genes (ISGs). Transcription of ISGs can also result from IFN-I induced homodimerization of STAT1 and its binding to another consensus sequence called gamma-activated sequence (GAS) situated in the promoter of ISGs. Finally, IFN-I can also induce a set of genes expression independent on STATs and involving MAPKs and PI3K pathway [20].

### 2.2. HIV Sensing by Pattern Recognition Receptors

#### 2.2.1. Sensing by Toll-Like Receptors

Both TLRs expressed on cell membranes and endosomal TLRs recognize HIV PAMPs and activate innate immune responses. The cell surface-expressed TLR2 and TLR4 are involved in the sensing of HIV-1 glycoprotein gp120, in particular in mucosal epithelial cells, leading to activation of NF-κB and production of inflammatory cytokines [21]. Even if mucosal epithelial cells are not themselves a target of HIV infection, they can sense extracellular HIV viruses through gp120 binding to TLR2 or TLR4 and secrete proinflammatory cytokines contributing to the recruitment and activation of nearby immune cells. In addition to TLR2 and TLR4, endosomal TLR7/8, recognizing guanosine- and uridine-rich single-stranded RNA (ssRNA), play an important role in HIV sensing in particular in phagocytic immune cells such as dendritic cells (DCs), plasmacytoid dendritic cells (pDCs), a subset of DCs specialized in IFN-I secretion, and macrophages. After endocytosis of incoming viruses by these cells, genomic viral RNAs are delivered onto endosomal TLR7/8 and lead to the secretion of IFN-α and inflammatory cytokines [22,23]. Activation of TLR7/8 does not require HIV replication showing that it is an incoming viral genome which is detected [24]. Immune responses induced by TLR7 and TLR8 after HIV sensing are dependent on the signal adaptor MyD88 [25] and also involve the transcription factor IRF7. Interestingly, IRF7 polymorphisms have been linked to a differential ability of patients to secrete IFN-α in response to HIV-1 [26].

In reply to TLR activation, HIV-1 can exploit some components of the innate immune response to promote its own replication. Indeed, activation of the transcription factor NF-κB following TLR activation is used by HIV-1 to initiate the transcription of the integrated provirus [27]. Moreover, despite the ability of TLRs to sense HIV PAMPs, innate immune response in the early events of HIV-1 infection is limited and some studies describe HIV-1 as a poor inducer of the IFN responses [28] suggesting that HIV-1 is able to escape from TLR signaling. In line with this, TLR dysfunctions in DCs have been observed in HIV-positive patients [29]. Moreover, Blanchet and colleagues have shown that in DCs, incoming HIV particles are engulfed into immunoamphisomes where TLRs can be activated but HIV-1 prevents the formation of these immunoamphisomes and the induction of associated-immunes responses by inhibiting autophagy [30].

Among TLRs, TLR10 is unique because its ligands and functions are poorly understood. Very recently, it was shown that TLR10 can sense HIV-1 proteins [31]. In particular, gp41 is recognized as a TLR10 ligand leading to the induction of IL-8 and NF-κB. However, in this case, sensing of HIV-1 does not lead to antiviral functions but rather enhances HIV infection [31].

#### 2.2.2. Sensing by Cytoplasmic Receptors

PRRs expressed in the cytoplasm generally play a crucial role in the detection of infections by RNA viruses since most of these viruses produce intermediates of replication into the cytoplasm. For instance, the dsRNA helicase RIG-I is a sensor of cytosolic HIV genomic RNA recognizing secondary structures and inducing expression of ISGs [32,33]. However, HIV-1 is able to escape from RIG-I detection by mediating its degradation through a mechanism involving the viral protease [33]. In addition to viral genome, viral mRNAs newly synthesized after viral integration could also be sensed by RIG-I in infected macrophages driving a strong expression of ISGs and favoring viral restriction in these cells [34].

Beyond the sensing of the viral genome, retroviral replication produces nucleic acid intermediates during reverse transcription such as ssDNA, RNA:DNA hybrids and dsDNA. The presence of dsDNA in the cytoplasm is a danger signal and is detected by the cyclic guanosine monophosphate (GMP)-adenosine monophosphate (AMP) synthase (cGAS) to induce IFN-β [35]. cGAS, which belongs to the nucleotidyltransferase family, produces cyclic GMP-AMP (cGAMP ) in response to binding to dsDNA. Then, cGAMP functions as a secondary messenger, binds to the endoplasmic reticulum protein STING and activates TBK1 and downstream IRF3 and IRF7, driving the production of IFN-I [36,37,38]. Moreover, cGAS-synthesized cGAMP can be transferred from producing cells to neighboring cells through gap junctions where it promotes STING activation and antiviral immunity [39]. cGAS is an important sensor of retroviral DNA allowing cytokines production in response to HIV, murine leukemia virus (MLV) and simian immunodeficiency virus (SIV) [40]. Inhibition of HIV reverse transcription prevents IFN-β production mediated by cGAS confirming that DNA products of the reverse transcription step serve as PAMPs for cGAS [40]. The sensing of HIV-1 DNA by cGAS occurs in cell lines or in monocyte-derived macrophages (MDM) and monocyte-derived dendritic cells (moDC) in which restriction of SAMHD1 (sterile alpha motif and histidine aspartate domain containing protein 1) has been abolished by a pre-treatment with the SIV protein Vpx [40]. Indeed, as it will be discussed below, DCs, that play a major role in the induction of immune responses, do not normally get activated and efficiently infected by HIV-1 due to SAMHD1-mediated impairment of reverse transcription preventing the production of HIV-1 dsDNA [4,41]. Quite the opposite, DCs are naturally infected and activated by HIV-2 since the HIV-2 protein Vpx leads to SAMHD1 degradation enabling viral replication and production of dsDNA recognized by cGAS [41]. Thus, the permissiveness of DCs toward HIV-2 versus HIV-1 allows a better recognition of HIV-2 PAMPs produced during infection that can drive the activation of DCs. However, the production of viral dsDNA is not sufficient for optimal activation of cGAS. Indeed, the viral capsid which remains associated with viral dsDNA up to the nucleus interior of infected cells [42,43] modulates the recognition of dsDNA by cGAS [44]. For instance, HIV-2 capsid is permissive to cGAS-mediated sensing during the early steps of infection whereas the HIV-1 capsid evades sensing of the dsDNA before integration even if Vpx is provided [41,44]. Actually, HIV-1 and HIV-2 capsids are directly recognized in the nucleus by the cellular protein NONO which is essential for the association of cGAS with dsDNA [45]. NONO binds more efficiently to HIV-2 capsid compared to HIV-1 capsid, contributing to better recognition of viral dsDNA and activation of immune responses [45].

In addition to cGAS, the nuclear and cytosolic protein IFI16 binds to DNA products of HIV reverse transcription, including truncation products, and activates immune responses through the adaptor STING, the protein kinase TBK1 and the transcription factors IRF3 and IRF7 [46].

#### 2.2.3. Activation of the Inflammasome

The inflammasome is a multicomponent protein complex composed of a sensor (NLRP3 (NOD-like receptor family, pyrin domain containing 3) or AIM2 (absent in melanoma-2) for example), an adaptor called ASC or PYCARD, and the caspase 1. Activation of the inflammasome after sensing of PAMPs leads to the cleavage and activation of the caspase 1, which cleaves pro-IL-1β and pro-IL-18 into active proinflammatory IL-1β and IL-18 cytokines. Then IL-1β initiates inflammatory cascades leading to cytokines production and recruitment of immune cells. HIV-1 infection triggers inflammasome induction, in particular activation of the NLRP3 inflammasome, in various cell types including monocytes, macrophages and CD4+ T cells [47,48,49]. In monocytes, IL-1β production requires both the activation of TLR8 and NLRP3 [47].

## 3. Restriction Factors Belonging to the Interferon Stimulated Genes Family

Sensing of HIV-1 infection by PRRs, and the subsequent production of IFN-I, induces the upregulation of antiviral ISGs (Figure 2). Even if, as intrinsic immune factors, restriction factors are usually expressed constitutively, most of them can be upregulated by IFN and thus can be classified into this ISG family.

### 3.1. IFITM

Interferon-inducible transmembrane (IFITMs) proteins are a family of small proteins that are involved in immunity and in the restriction of many viral infections. The human IFITM family comprises of five members, including immune-related IFITM1, IFITM2 and IFITM3, as well as IFITM5 and IFITM10 which have no known role in immunity. IFITM proteins are type II transmembrane proteins [50] that were identified more than 30 years ago, but their antiviral function was described for the first time in 2009 thanks to an RNA interference screen for factors modulating influenza A virus infection [51]. IFITMs restrict a great number of enveloped RNA viruses including: influenza A virus [51], flaviviruses (Dengue virus, West Nile virus [51,52] and Zika virus [53]), Hepatitis C virus (HCV) [54], filoviruses (Marburg and Ebola viruses) [55,56], viruses from the bunyavirales order (Rift Valley fever virus [57], Hantaan virus [58]), coronaviruses (such as SARS-CoV) [55] and HIV-1 [9]. IFITM3 was also described to restrict infection by non-enveloped RNA viruses (reovirus [59]) and enveloped DNA viruses (vaccinia virus [60]). A very recent study by Smith and colleagues has enlarged the number of viruses affected by IFITMs [61]. The authors showed that IFITM1 restricts replication of several other RNA viruses such as Paramyxoviridae and Pneumoviridae (respiratory syncytial virus (RSV), mumps virus and human metapneumovirus (HMPV)) and DNA viruses (herpes simplex virus 1 (HSV-1)) which both share the feature of entering cells via the plasma membrane [61].

IFITMs that localize at the plasma membrane as well as the membranes of endocytic vesicles and lysosomes [62,63], restrict viral infections by inhibiting the entry step. It has been described that IFITMs block viral entry by impairing the hemifusion process most likely by reducing membrane fluidity [64]. The potency of IFITM restriction of viral cell entry is generally correlated to the colocalization of IFITMs at the sites of viral fusion. Inhibition of virus–cell fusion does not block endocytosis of virions into cells, allowing for them to target endosomal or lysosomal compartments [65,66,67,68]. Thus, the first antiviral action of IFITMs is to protect target cells from incoming viruses (Figure 2). In addition to this protective function, a second antiviral mechanism of IFITMs (particularly IFITM2 and 3) named “negative imprinting of virions”, leads to the production of virions that package IFITMs and display reduced entry into target cells [66,69]. During viral assembly into infected cells, IFITMs are incorporated into nascent HIV particles leading to a decrease in the virions infectivity. Fusogenic properties of negative imprinted virions are reduced, thus preventing their fusion with new target cells [66,69]. A study by Yu and colleagues explains the impairment of HIV fusogenic property by a decrease in the number of envelope spikes in virions [70]. However, Tartour and colleagues showed that the reduction of virion infectivity by IFITMs is not restricted to HIV-1 but is generalizable to other viruses belonging to different genus [71]. Thus, the negative imprinting of viral particles seems to be a conserved antiviral property of IFITMs. Very recently, the regions of IFITM3 involved in the negative imprinting of HIV virions have been mapped [72].

The human IFITM locus is located on chromosome 11 and composed of five genes: IFITM1, IFITM2, IFITM3, IFITM5 and IFITM10. IFITM1, IFITM2, and IFITM3 are expressed nearly ubiquitously in humans, whereas IFITM5 is expressed primarily in osteoblasts. IFITM1 (also known as Leu 13 antigen) was described to be induced by IFN-α and the type II IFN-γ [73]. More generally, as ISGs, IFITM1, IFITM2, IFITM3 genes each have an interferon-stimulated response element (ISRE) in its promoter region besides the additional gamma-activated sequence (GAS) in the promoter region of IFITM1 gene [74]. The IFITM3 promoter has binding sites for several transcription factors activated by innate immune signaling pathways such as IRF1, STAT1, TBP, STAT3, STAT2 [75]. Thus, IFITMs are upregulated by innate immunity which enhances their antiviral functions (for review see [76,77]). Importantly, despite their name, IFITM5 and IFITM10 are not induced by IFNs [78].

### 3.2. TRIM5α

Tripartite motif (TRIM) proteins constitute a large family of E3 ubiquitin ligases involved in many cellular functions such as cell differentiation, apoptosis, autophagy, immunity and antiviral functions [79,80,81,82]. TRIM proteins share a conserved N-terminal RBCC structure that is characterized by a RING finger E3 ligase domain (R), one or two B-box domains (B), a coiled-coil domain (CC) and a variable C-terminal domain that defines the subgroup of TRIM family members. The C-terminal domain of TRIM5α is the SPRY-PRY domain which is the most frequent among TRIM proteins. Several TRIM proteins have antiretroviral activity [82,83] and among them, TRIM5α was identified as the factor responsible for HIV-1 restriction in rhesus macaques [8] and restriction of several other mammalian viruses such as MLV [84,85,86,87]. TRIM5α is a cytoplasmic protein that binds to viral capsids and inhibits lentiviruses replication at an early step after entry [88,89]. Viral restriction mediated by TRIM5α is host and virus-specific due to a considerable inter and intra-species variation in TRIM5α sequence [90,91]. TRIM5α proteins from Old World monkeys restrict a broad range of retroviruses including HIV-1, N-tropic murine leukemia virus (N-MLV) and equine infectious anemia virus (EIAV), but are not efficient against some SIV strains [8,87]. Yet in New World monkeys, TRIM5α proteins do not generally restrict HIV-1. The human orthologue (hTRIM5α) strongly inhibits N-MLV and EIAV but is unable to efficiently restrict HIV-1 [8,84,85,87]. However, Ribeiro and colleagues have shown that hTRIM5α can inhibit HIV-1 infection in some DC subsets but this antiviral function depends on the route of HIV-1 internalization. C-type lectin receptor-dependent uptake of HIV-1 leads to hTRIM5α restriction whereas internalization via DC-SIGN (Dendritic Cell-Specific Intercellular adhesion molecule-3-Grabbing Non-integrin) binding does not [92].

The C-terminal domain SPRY-PRY is crucial for the antiviral activity of TRIM5α because it is responsible for the binding to retroviral capsids [88,89,93,94]. The binding of TRIM5α to viral capsid promotes premature disassembly of the viral capsid, preventing reverse transcription and further integration (Figure 2) [93,95]. Comparison of the rhesus macaque TRIM5α sequence with hTRIM5α identified a single amino acid position within the SPRY-PRY domain which is crucial for binding of HIV-capsid [96,97]. Restriction by TRIM5α is determined by the sequence of the viral capsid that allows the binding or not [94,98]. Thus, HIV-1 capsid could have evolved to escape from hTRIM5α recognition [99].

While the SPRY-PRY domain of TRIM5α is responsible for the recognition of viral capsid, the E3 ubiquitin ligase RING domain is important for effector function probably through the recruitment of the proteasomal machinery [100]. Indeed, inhibition of proteasome prevents the premature disassembly of the capsid and restores HIV-1 reverse transcription [101,102,103].

TRIM5α is constitutively expressed, but several studies have shown that its expression is upregulated by IFN-I. Indeed, treatment of HeLa and HepG2 cells with IFN-I increases TRIM5α promoter activity and mRNA levels [104]. In primary cells, IFN-I enhances TRIM5α expression in humans, African green monkeys and macaque cells [105]. IFN-mediated upregulation of TRIM5α in rhesus monkey cells enhances HIV-1 replication blocking [106]. In human cells, IFN-α treatment results in enhanced antiviral activity against N-MLV infection [105,106]. A systematic analysis of TRIM genes expression in human primary lymphocytes and monocyte-derived macrophages in response to IFN-I and IFN-II has revealed that several TRIM genes are upregulated by IFN [107]. Among the 72 human TRIM genes tested, 27 were sensitive to IFN including TRIM5α. TRIM5α has no homologue in a mouse model, but an analysis of mRNA expression of other TRIM proteins in mice macrophages, myeloid and plasmacytoid dendritic cells, and a selection of CD4+ T cell subsets showed that a lot of mice homologues of human TRIM are upregulated by IFN [108]. Taken together, TRIM proteins can be considered as innate immunity-induced factors contributing to the establishment of an antiviral state even if HIV-1 has evolved to escape from hTRIM5α recognition.

### 3.3. APOBEC3G

APOBEC3G (A3G) proteins or apolipoprotein B mRNA-editing enzyme catalytic subunit-like 3G, also known as CEM15, belong to the APOBEC family of cytidine deaminase enzymes including seven members (A, B, C, D, F, G and H). A3G was the first restriction factor identified for HIV-1. It was the interaction of the HIV-1 virion infectivity factor (Vif) with A3G that led to its identification as a restriction factor [109]. Indeed, whereas Vif-deleted HIV-1 viruses replicate in permissive T cell lines that do not express A3G, Vif is required for HIV-1 replication in non-permissive cells such as CD4+ T cells and monocyte-derived macrophages expressing A3G [109,110]. Vif mediates the proteasomal degradation of A3G preventing its antiviral function [111]. When A3G is not degraded by Vif, its antiviral activity occurs after the first cycle of infection. A3G proteins are encapsidated into budding HIV virions due to their ability to interact with the viral nucleocapsid protein and with nucleic acids [112]. In newly infected cells, the antiviral effect of A3G is exerted during reverse transcription (Figure 2). The deaminase activity of the enzyme converts cytosines to uracils in the (−) strand DNA leading to a massive G-to-A mutation of the nascent viral DNA genome. Hypermutated proviral DNA can be correctly integrated into the host genome but A-to-G hypermutations introduce amino acid substitutions and premature STOP codons preventing translation of viral messengers [2,3,113,114,115]. Mutations can also target regulatory portions of viral DNA such as the trans-activation response (TAR) element affecting the transcriptional activity of HIV [116]. It has also been proposed that the degradation of deaminated viral DNA by the uracil base excision repair (BER) pathway could take part in the antiviral activity of A3G, but the exact contribution of this pathway in HIV restriction is controversial [117,118,119,120]. Furthermore, the deaminase activity of A3G is not absolutely required for antiviral activity since A3G can also block reverse transcription elongation in a deaminase-independent manner [121,122,123,124,125]. Antiviral activity of APOBEC3 proteins is not restricted to HIV-1 since these proteins affect a broad range of viruses including Hepatitis B virus (HBV) [126] and other retroviruses, SIV, HTLV-1 [127], foamy virus [128], MLV [129] and mouse mammary tumor virus (MMTV) [130].

A3G protein, which is the most abundant protein of the APOBEC3 family, is expressed in multiple cells with particularly high level expression in immune cells such as activated T cells, monocytes, macrophages and mature DCs [131,132]. Activation of innate immunity, in particular the type I IFN response, leads to an increase in A3G expression. IFN-α was the first cytokine identified as a strong inducer of A3G expression. Peng and colleagues showed that IFN-α has the potential to promote A3G and to override HIV Vif neutralization in macrophages. Even if IFN-α induces its antiviral activity through multiple mechanisms including PKR and RNase L, the silencing of A3G by siRNA partially prevented the inhibition of HIV suggesting that A3G is a key downstream anti-HIV mechanism induced by IFN-α [133]. In macrophages, IFN-γ can also increase A3G expression but to a lesser extent than IFN-α [133,134]. Regulation of A3G expression by IFN-α is cell-type dependent. For example, it was shown that A3G was upregulated by IFN-α in liver cells and macrophages, but not in T lymphoid cells or epithelial 293T, whereas other IFN-α-stimulated genes such as dsRNA-activated PKR were induced in all these cells, suggesting that additional cellular factors may regulate IFN-α-induced A3G expression [135]. Regarding CD4+ T lymphocytes, the main targets of HIV infection, IFN-α enhances the expression of A3G in human primary resting but not in activated CD4+ T cells. In resting CD4+ T cells, A3G expression is associated with an inhibition of HIV-1 replication at its early stage [136]. Analysis of A3G, A3F and BST-2 expression levels in HIV/HCV coinfected, antiretroviral therapy-naïve individuals before, during and after pegylated IFN-α/ribavirin combination therapy showed that A3G/3F and BST-2 mRNA expressions were significantly elevated during IFN-α/riba treatment in patient-derived CD4+ T cells [137]. IFN-α, endogenously produced by innate immunity activation or exogenously added, also induced expression of A3G in pDCs [138] and in immature moDCs [139] leading to HIV restriction. Interestingly, both IFN-α and IFN-γ significantly enhance the expression of A3G and drastically inhibit HIV-1 replication in primary human brain microvascular endothelial cells (MVECS), the major component of the blood–brain barrier suggesting that IFN-induced A3G pathway is involved in the protection of the central nervous system from HIV invasion [140].

As an ISG, A3G expression is induced by PRRs after the detection of PAMPs. For example, treatment of immature DCs or monocyte-derived macrophages, with polyI:C, a synthetic dsRNA recognized by TLR3, leads to A3G induction and antiviral activity against HIV [141,142]. The stimulation of human moDCs with Gag virus-like particles (VLPs), which are recognized by PRRs, leads to the production of IFN-α, along with an increase in mRNA and protein expression of A3G. The stimulation of innate immunity and restriction factor A3G by Gag-VLPs drag an antiviral state that inhibits a later HIV-1 infection. Both the increase in A3G expression and the inhibition of HIV-1 replication were reverted by anti-IFN-α or anti-IFNAR antibodies [143]. Furthermore, several cytokines involved in innate and adaptive immunity such as IL-2, IL-7 and IL-15 induce A3G expression in peripheral blood lymphocytes demonstrating that innate immunity and the restriction factor A3G are intimately linked [144].

### 3.4. SAMHD1

SAMHDI (sterile alpha motif and histidine aspartate domain containing protein 1) is a deoxynucleoside triphosphate phosphohydrolase (dNTPase) that impairs HIV reverse transcription by decreasing the pool of cellular dNTPs, and as a consequence, the pool of nucleotides available for reverse transcription (Figure 2) [145,146,147]. SAMHD1 hydrolyzes all four dNTPs to deoxynucleosides and inorganic triphosphates. As a consequence, SAMHD1 inhibits HIV-1 replication in myeloid cells including moDCs and monocyte-derived macrophages [4,5]. HIV-1 infection is also restricted by SAMHD1 in resting CD4+ T cells but not in activated T cells [148,149]. The low efficiency of SAMHD1 restriction in dividing cells can be explained by the regulation of SAMHD1 activity at the post-transcriptional level and in a cell-cycle-dependent manner [150,151,152]. Indeed, SAMHD1 activity seems to be controlled by cyclin-dependent kinases (CDK), CDK1, CDK2 and CDK6 which are only active in dividing cells [153,154,155]. The mechanism of SAMHD1 regulation by CDK is not fully understood. Phosphorylation of SAMHD1 at threonine 592 could interfere with the stability of long-lived stable SAMHD1 tetramers, which thereby lose their ability to efficiently bind and degrade dNTPs resulting in the increase of intracellular dNTPs [156,157]. Alternatively, T592 phosphorylation might also down-modulate SAMHD1 dNTPase function without fully inhibiting the activity of the enzyme [154]. However, some studies have reported that T592 phosphorylation does not affect the dNTPase activity of the enzyme suggesting that another mechanism independent of dNTPase activity is involved in SAMHD1-mediated viral restriction [156,158]. For review, see [159]. Whatever the mechanism involved, T592 phosphorylation of SAMHD1 severely affects HIV-1 restriction [156,160]. Stimulation of resting CD4+ T cells (in which SAMHD1 is active) with IL-7 induces T592 phosphorylation of SAMHD1 abrogating its antiviral activity [161]. Interestingly, SAMHD1 also binds ssDNA and RNA and was described to be able to degrade RNA–DNA duplexes [162,163] as well as HIV genomic RNA [164,165]. Using several SAMHD1 mutants that retain only one of the two enzymatic activities, either RNase or dNTPase activity, Ryoo and colleagues provided evidence that the RNAse activity of SAMHD1 has a crucial role in HIV-1 restriction [164]. However, others did not find involvement of RNAse activity of SAMHD1 in viral restriction [166], rendering the contribution of this activity as an open question.

The antiviral activity of SAMHD1 is not restricted to HIV-1 since several other retroviruses including bovine, feline and simian immunodeficiency viruses [167,168], HTLV-1 [169] and dsDNA viruses (vaccinia virus and HSV) [170] are targets of SAMHD1. In contrast, HIV-2 is not affected by SAMHD1 restriction because the accessory protein Vpx induces SAMHD1 degradation [4,41]. Vpx counteraction of SAMHD1 explains why DCs are productively infected by HIV-2 whereas they are not efficiently infected by HIV-1.

SAMHD1 is not described as an ISG since its expression is not induced by IFN-I in DCs and primary CD4+ T cells [171]. However, activations of TLR3 and RIG-I/MDA5 using agonists were shown to upregulate SAMHD1 expression in HeLa cells, HEK293 cells and MARC-145 cells [172]. In this model, the increase of SAMHD1 expression is mediated by the transcription factor IRF3 but is independent of the classical IFN-induced JAK-STAT pathway. In addition, IFN-α treatment induces SAMHD1 expression in liver cells in a STAT1, STAT2 and IRF9 dependent pathway [173,174]. In monocyte-derived macrophages, IL-12 and IL-18 treatments lead to the overexpression of SAMHD1 which renders cells resistant to HIV-1 infection [175]. All together, these studies suggest that innate immune response mediators could induce SAMHD1 to promote its antiviral activity.

### 3.5. Mx2

The Myxovirus resistance protein 2 (Mx2 or MxB) is a protein belonging to the dynamin superfamily of large guanosine triphosphatases (GTPases) [176] and is part of ISG. Both IFN-α and IFN-β induce Mx2 expression in human fibroblasts with an accumulation of Mx2 proteins in the cytoplasm [177]. Cloning and sequence analyses of the cDNA coding Mx2 identified five regulatory regions in the promoter conferring IFN-I inducibility [178]. In human primary cells, Mx2 proteins are detected in monocytes and lymphocytes following IFN-α treatment [179]. In addition, Mx2 is also induced by IFN-γ [180] which leads to the suppression of HIV infection in macrophages [181]. Interestingly, Mx2 was identified as a restriction factor inhibiting HIV in screens aiming for identifying cellular factors responsible for the inhibitory action of IFN-α at early steps of the HIV-1 replication cycle [10,182]. Mx2 was shown to inhibit the nuclear accumulation and integration of HIV-1 reverse transcripts by a mechanism involving the capsid of HIV [10]. The reverse transcription step is not affected by Mx2 but 2-long terminal repeat circular forms of non-integrated HIV-1 DNA are less abundant in Mx2-expressing cells, suggesting that Mx2 could inhibit HIV-1 nuclear import or could destabilize nuclear HIV-1 DNA [182]. Thus, Mx2 could affect nuclear entry or post-nuclear trafficking without destabilizing the inherent catalytic activity of viral preintegration complexes [183]. Some studies also suggest that Mx2 could impair HIV uncoating by binding to the capsid [184]. Interaction between Mx2 and the viral capsid is either direct [185] or mediated via the interaction with CypA [186]. Oligomerization of Mx2 seems to be required for its antiviral activity since monomeric Mx2 is not antiviral [187,188]. The GTPase activity of Mx2 is required for oligomer assembly [189] even if this activity has been dispensable for its restriction activity [183]. Very recently, Buffone and colleagues showed that Mx2 could be involved in HIV restriction mediated by SAMHD1 [190]. Indeed, Mx2 knock-out in THP-1 or primary macrophages renders SAMHD1 ineffective against HIV-1. However, the mechanism involved is not identified since Mx2 does not affect the cellular localization of SAMHD1, nor its phosphorylation or the ability of SAMHD1 to decrease cellular levels of dNTPs [190]. Therefore, the involvement of Mx2 in SAMHD1-mediated restriction needs to be further investigated.

The identification of Mx2 as a factor mediating the antiviral function of IFN-α in HIV infection suggests that this restriction factor could play an important role in vivo. Interestingly, human population genetic analyses identified that the ancestral allele of Mx2 protects from HIV-1 infection and is associated with lower in vitro HIV-1 replication and increased Mx2 expression in response to IFN-α [191]. In addition to HIV, Mx2 is also able to restrict infections by other human viruses such as HSV-1 and 2 and Kaposi’s sarcoma-associated herpesvirus (KSHV) in the context of IFN-I [192] and Hantaan virus in response to IFN-λ [193].

### 3.6. Tetherin/BST-2

Tetherin/BST-2 (bone marrow stromal antigen 2) is a restriction factor that impairs HIV-1 particles budding. Before its identification, it was known that in human cells proteins-based tethers (named tetherins) retain the fully formed virions on infected cell surfaces, preventing the release of HIV-1 particles. Moreover, the HIV-1 protein Vpu has been described as an inhibitor of this antiviral activity. In 2008, BST-2 was identified as the tetherin responsible for this restriction activity and blocked by Vpu [6,7]. BST-2 is a transmembrane protein with both a N-terminal transmembrane domain and a C-terminal glycosyl-phosphatidylinositol (GPI) anchor [194]. Both these two membrane anchors can interact with the membrane of virions allowing the incorporation of BST-2 in HIV-1 budding particles [195,196]. It is the unique topology of BST-2, rather than the primary sequence, that is critical for its antiviral activity [195]. Immunoelectron microscopy experiments revealed that BST-2 forms a bridge between viral particles and cellular plasma membranes causing the retention of the nascent viral particles at the surface of infected cells (Figure 2) [197]. Linear filamentous strands enriched in BST-2 can be observed between patches of virions and the plasma membrane. Venkatesh and colleagues proposed several models of the configuration adopted by BST-2 during virion entrapment [198]. However, their results suggest that BST-2 dimers could adopt a configuration in which pairs of N-termini transmembrane domains or pairs of GPI-anchored-C-termini are inserted into assembling virion particles (with a preference for the insertion of GPI-anchored-C-termini), while the remaining pair of membrane anchors remains embedded in the infected cell membrane [198]. In addition to preventing the release of budding virions, BST-2 can also target them for internalization and degradation via the endosomal/lysosomal pathway [7,195,199]. Two-isoforms of BST-2 co-exist in cells: a long isoform and a shorter isoform lacking the first 12–17 amino acids. Both isoforms can form homo- and heterodimers and have an antiviral activity [200]. However, the shorter isoform is more resistant to Vpu counteraction.

BST-2 is expressed at different levels in a variety of human tissues. For example, BST-2 is highly expressed on blood vessels whereas the expression on pDCs is not constitutive [201]. The promoter region of the BST-2 gene has a binding site for the transcription factor STAT3 [202] suggesting that BST-2 expression can be upregulated by innate immune signaling pathways such as IFN-I. Indeed, a lot of studies relate to the induction of BST-2 expression by IFN-I and IFN-II in several human cell types. For example, BST-2 expression is upregulated by IFN-γ in Human Umbilical Vein Endothelial Cells (HUVEC) [203], by the three types of IFN (IFN-α, IFN-γ and IFN-λ) in hepatocytes [204] and by IFN-I in neurons via a STAT1 dependent pathway [205]. BST-2 is also overexpressed in myeloid DCs (myDC) and moDCs by IFN-α as well as TLR4 engagement with lipopolysaccharides (LPS) [206]. Very interestingly, LPS, but not IFN-α stimulation of immature DCs, leads to a redistribution of BST-2 at the virological synapse between DCs and CD4+ T cells exacerbating HIV-1 restriction. TLR3 engagement by polyI:C in macrophages also increases BST-2 expression [142]. Taken together, these data show that BST-2 is a gene induced by innate immunity after the sensing of infection by TLRs. This is confirmed by in vivo studies showing that HIV infection upregulates BST-2 expression in human peripheral blood mononuclear cells (PBMC). Comparing BST-2 expression in PBMC from healthy donors or HIV-positive patients, Homann and colleagues showed that BST-2 increases as a result of HIV infection, in particular during the acute infection phase [207]. Treatment of PBMC in vitro with IFN-α or TLR agonists increased the expression of BST-2 to levels similar to those found during infection in vivo. Moreover, BST-2 can also be induced by IL-27 in human monocytes and T cells [208].

Mechanistically, upregulation of BST-2 expression by IFN-I seems to be dependent on STAT1 phosphorylation since, in cell lines deficient for STAT1 or unable to phosphorylate STAT1, this upregulation is lost [209]. The dependence on STAT1 for BST-2 upregulation by IFN-I suggests that BST-2 behaves as a classical ISG. Indeed, the first step of IFN-I cascade after the binding of IFNAR receptors involves the phosphorylation of STAT1, which is required to form the ISGF3 complex that recognizes ISRE sites in the promoter of ISGs. However, mutational analysis of the BST-2 promoter showed that STAT and ISRE binding sites were not required for induction by IFN-I suggesting that the ISGF3 complex may not be acting directly on the BST-2 promoter but that a factor regulated by ISGF3 is required for BST-2 regulation [209]. This study of Bego and colleagues evidences that IRFs (in particular IRF-1, IRF-3 and IRF-7) are the IFN-induced gene products required for activation of the BST-2 promoter following IFN signaling. Nevertheless, since these data mainly originate from experiments on cell lines, it would be interesting to confirm if IRFs can induce BST-2 upregulation in primary cells and in vivo. Indeed, since IRFs are induced following TLR activation by viral infections, the upregulation of BST-2 expression in virus-infected cells in response to virus-induced IRF-7 activation would ensure that host cells maintain BST-2-mediated virus restriction. Thus, the induction of BST-2 could be an integral part of innate immune responses.

### 3.7. Other Interferon Stimulated Genes Restricting HIV Infection

#### 3.7.1. Cholesterol-25-Hydroxylase

Cholesterol 25-hydroxylase (CH25H) is an enzyme converting cholesterol to 25-hydroxycholesterol (25-HC). In mice macrophages and in bone marrow-derived dendritic cells (BMDC), Ch25h expression is induced after TLR4 stimulation with LPS or TLR3 stimulation with polyI:C [210,211]. The upregulation of Ch25h transcription after TLR activation is dependent on the production of IFN-I by a TRIF-mediated pathway. Treatment of BMDCs and macrophages with IFN-α or IFN-β induces Ch25h expression in wild type (WT) mice but not in *Ifnar-/-* mice demonstrating that IFN-I signaling through IFNAR is required [211]. IFN-γ also induced Ch25h expression showing that this enzyme is regulated by several types of IFN. Additional studies on human cells would be interesting to confirm the regulation of Ch25h expression by IFN in humans. The antiviral function of CH25H was described by Liu and colleagues in a work demonstrating that treatment of cultured cells with 25-HC broadly inhibits replication of several enveloped viruses, including HIV, by impairing fusion between the viral envelope and cell membrane [212]. The antiviral function of CH25H has also been described for HCV and the Zika virus infection [213,214]. The mechanism by which CH25H inhibits viral entry was recently described in more detail [215]. Using biophysical approaches, Gomes and colleagues have shown that the conversion of cholesterol in 25-HC by CH25H alters the fluidity of lipid membranes and decreases the conformal plasticity of HIV fusion peptide preventing the formation of the fusion pore and resulting in the blocking of virus–cell fusion [215]. Thus, CH25H is a part of ISG induced by innate immunity which could contribute to the establishment of an antiviral state. However, more studies will be helpful to determine the relevance of CH25H antiviral function in vivo.

#### 3.7.2. Zinc-Finger Antiviral Protein (ZAP)

ZAP is an IFN-inducible protein encoded by the ISG ZC3HAV1 and exerting antiviral activity against a broad range of viruses including retroviruses [216], alphaviruses [217], filoviruses [218], HBV [219], coxsackievirus B3 [220] and Japanese encephalitis virus [221]. ZAP was first discovered to inhibit Moloney murine leukemia virus (MMLV) by leading to the loss of viral mRNAs in the cytoplasm but not in the nucleus of infected cells [216]. ZAP is able to directly bind to mRNAs through its CCCH zinc finger motifs [222] and recruits the RNA exosome to degrade the target RNA [223]. ZAP also interacts with the p72 DEAD-box RNA helicase [224] and through this interaction, recruits the decapping complex Dcp1a/Dcp2 to initiate degradation of the target viral mRNA from the 5’ end [225]. ZAP can mediate the degradation of viral mRNAs through an additional mechanism. In the case of HIV, in addition to the recruitment of p72, ZAP selectively recruits cellular poly(A)-specific ribonuclease (PARN) which removes the poly(A) tail of target viral mRNA and recruits the RNA exosome to degrade the RNA body from the 3′ end [225]. By this way, ZAP specifically targets the multiply spliced but not unspliced or singly spliced HIV-1 mRNAs for degradation. A third antiviral mechanism of ZAP is inhibition of translation. Indeed, ZAP can prevent the interaction between translational initiation factors eIF4G and eIF4A to block the translation of viral mRNA [226]. Recent studies report that the ubiquitin E3 ligase TRIM25 is required for the antiviral activity of ZAP because it modulates the target RNA binding activity of ZAP [227,228].

As an ISG, ZAP expression is upregulated in human cells by IFN-α treatment [219]. Two isoforms of ZAP, ZAP-L (902 aa) and ZAP-S (699 aa), that differ at their C-termini due to alternative splice variants have been described. ZAP-L was shown to have greater antiviral activity than ZAP-S [229] whereas ZAP-S was induced to a greater extent after IFN treatment [219,230]. Very recently, two additional splice variants of human ZAP, ZAP-XL (extra-long) and ZAP-M (medium) have been identified [231]. These isoforms show different antiviral activities depending on the virus targeted. For instance, the longer ZAP isoforms better inhibit alphaviruses and HBV while all isoforms equally inhibit Ebola virus [231].

In conclusion, ZAP is an IFN-inducible protein contributing to HIV restriction in infected cells. Interestingly, ZAP is negatively regulated by Matrin 3, a protein containing two RNA recognition motifs and acting as an HIV-Rev cofactor [232,233]. Thus, HIV could limit ZAP antiviral functions by using Matrin 3. In agreement with this hypothesis, when Matrin 3 is knocked-down, ZAP action is exerted on both unspliced and multiply-spliced HIV-1 transcripts whereas it is normally restricted to multiply-spliced mRNA [232].

#### 3.7.3. Schlafen 11

Schlafen genes are a family of ISGs that are induced following activation of the IRF3 pathway. Among this family, Schlafen 11 (SLFN11) has been identified as an inhibitor of HIV replication [12]. SLFN11 impairs HIV translation by preventing change in the composition of the tRNA pool induced by HIV infection. Due to the high frequency of A/T nucleotides in HIV codons compared to cellular codons, the translation of viral mRNA is not optimal in physiological conditions. To overpass this limitation to replication, HIV induces a change in the composition of the tRNA pool, increasing the frequency of tRNA decoding A-ending codons [234]. By preventing this change, SLFN11 selectively inhibits viral protein synthesis. Translation shut-off is a general antiviral mechanism that is mediated by other ISGs such as the protein kinase R (PKR) that phosphorylates the eIF2α factor. However, in the case of PKR, both cellular and viral translation are impaired. In contrast, translation inhibition mediated by SLFN11 could be more retrovirus specific since it affects codons enriched in A/T nucleotides. The role of SLFN11 in counteracting retroviruses is leant by the fact that equine SLFN11 restricts EIAV via a mechanism targeting translation similar to the one used by human SLFN11 [235]. Moreover, the function of SLFN11 is conserved among primates since chimpanzee, orangutan, gibbon and marmoset encode a SLFN11 protein with a higher activity compared to human SLFN11 [236].

#### 3.7.4. ISG15

Interferon-stimulated gene 15 (ISG15) is a 15 kDa protein that belongs to the family of ubiquitin-like modifiers and is induced by IFN-α and IFN-β as well as by activation of the NF-κB pathway [237]. ISG15 expression can also be induced by viral infection and dsRNA [238]. ISG15 is present in cells in a soluble form or it is conjugated to target proteins via enzymatic reactions similar to the ubiquitin conjugation pathway [239]. The conjugation of ISG15 on target is called ISGylation. ISG15 impairs the replication of several viruses including HIV-1 [240]. Antiviral functions can be exerted through ISGylation or by free ISG15. For example, free ISG15 is able to inhibit ubiquitination by negative regulation of Nedd4, a cellular protein involved in the ubiquitination of viral proteins [241]. In the case of HIV-1, the restriction function of ISG15 seems to be linked to ISGylation which is mediated by HERC5 ( HECT and RLD domain containing E3 ubiquitin protein ligase 5), a type E3 protein ligase [242]. HERC5-mediated ISGylation results in the accumulation of Gag at the plasma membrane demonstrating a defect in HIV particles release [242,243]. In fact, ISGylation inhibits Gag and Tsg101 ubiquitination disrupting the interaction between these two proteins. Tsg101 is a cellular protein of the endosomal sorting complexes required for transport (ESCRT) pathway that is essential for HIV budding so the disruption of Gag and Tsg101 interaction prevents HIV release [240,244]. Interestingly, a study on HIV-positive patients has shown that ISG15 is one of the highest ISG expressed [245]. Indeed, ISG15 expression was increased in untreated HIV-1 patients compared to healthy donors and the amount of ISG15 correlated with viral load [245]. However, it was also shown that HIV-1 infection of THP-1 cells reduces the IFN-α-mediated induction of ISG15 [246]. Similarly, PBMCs from HIV-infected patients show a reduction in IFN-α-induced ISG15 compared to healthy controls suggesting that HIV-1 may have developed strategies to escape from ISG15 antiviral function.

#### 3.7.5. Guanylate-Binding Protein 5 (GBP5)

Guanylate-binding proteins belong to the family of interferon-inducible GTPases and are involved in cell-intrinsic immunity against bacteria and viruses [247]. Human GBP are composed of seven members (GBP1–GBP7) located on chromosome 1. Among GBP, GBP5 has been identified to reduce infectious HIV-1 production in transfected 293T [248]. In fact, GBP5 is expressed in macrophages and CD4+ T cells in which it plays an antiviral function by impairing the incorporation of HIV-gp120 in newly produced virions decreasing viral progeny infectivity [248,249,250]. This antiviral activity does not involve the GTPase function of GBP5 but requires the C-terminal isoprenylation domain of GBP5, mediating the anchoring to Golgi membranes, where env processing occurs. Misincorporation of gp120 is due to a defect in env glycosylation impairing the processing of gp160 precursor [249]. GBP5 is induced by IFN-γ, in particular in macrophages [247], to the extent that its expression is a marker of IFN-γ-mediated macrophage activation [251]. However, HIV may evade GBP5 antiviral activity through mutations enhancing the translation of env proteins. For instance, since vpu and env are expressed from the same bicistronic RNA, some mutations in the vpu gene disrupting vpu reading frame increase env expression and confer resistance to GBP5 [249]. It is interesting to note that M-tropic HIV-1 strains show high frequencies of defective vpu gene, probably resulting from a selection allowing the virus to escape from antiviral activity of GBP5 [252].

## 4. Restriction Factors Shaping Immunity

### 4.1. Restriction Factors Promoting Immunity

#### 4.1.1. IFITM

IFITMs are induced by innate immunity, but very few studies have investigated if IFITMs could, in return, shape immune responses. It has been shown, in a mouse model of colitis, that IFITM3 knock-out leads to impaired anti-inflammatory cytokine expression exacerbating inflammation [253]. This study suggests that IFITM genes could be involved in the regulation of inflammation. Another study in mice showed that IFITMs are involved in adaptive immunity and influence Th1/Th2 polarization of CD4+ T cells [254]. In the case of viral infections such as HIV, it would be interesting to study if IFITMs promote innate immunity in addition to their antiviral functions. Indeed, since IFITMs are localized in endosomes [63] where the immune sensors TLRs are expressed, it is tempting to speculate that they might be involved in targeting viruses for TLR sensing (Figure 2).

#### 4.1.2. TRIM5α

Due to its E3 ubiquitin ligase domain, TRIM5α is involved in signal transduction, particularly in innate immune signaling [255]. TRIM5α catalyzes the synthesis of unanchored K63-linked ubiquitin chains that activate the TAK1 kinase complex leading to downstream stimulation of AP-1 and NFκB signaling [89]. Interaction with the viral capsid lattice greatly enhances the E3 activity of TRIM5α and induces the transcription of AP-1 and NF-κB-dependent factors with a magnitude that tracks with TRIM5α avidity for the invading capsid [89,256]. The synthesis of free K63-linked ubiquitin chains that is necessary for activation of both NF-κB and AP-1 is dependent on a conserved sumoylation consensus site present between the RING motif and the N-terminal extremity of TRIM5α [257,258]. Involvement of TRIM5α as a positive regulator of innate immunity is a common feature within the TRIM family since half of known human TRIMs enhance innate immunity [259] with 16 TRIM proteins able to induce NF-κB and/or AP-1 [260]. Very interestingly, in DCs, endogenous TRIM5α accumulates in nuclear bodies in a SUMOylation-dependent manner and this leads to potent induction of IFN-I mediated by the sensing of viral reverse-transcribed DNA by cGAS [261]. Taken together, these data show that TRIM5α can be considered as a sensor of the viral capsid that induces innate immune signaling as other PRRs. Recently the protein NONO has been identified as a sensor of HIV-2 capsid allowing cGAS association with HIV DNA in the nucleus for induction of innate immune responses [45]. Since reverse transcribed genomes reach the nucleus in association with the capsid, the presence of a capsid sensor in the nucleus is of great interest for the detection of infection. This highlights why HIV-1 needed to evolve in order to escape hTRIM5α recognition. Restriction of HIV-1 by rhesus TRIM5α shows the strong potential of TRIM5α as a viral sensor inducing antiviral functions (Figure 2). Moreover, rhesus TRIM5α has been described to promote adaptive immunity by enhancing the recognition and killing of HIV-1-infected cells by CD8 T cells [262].

#### 4.1.3. APOBEC3G

Beyond its antiviral function, the deaminase activity of A3G can shape innate and adaptive immunity. Norman and colleagues have shown that cytidine deamination of HIV viral DNA by A3G enhances the recognition of HIV-infected cells by natural killer cells (NK) [263]. Indeed, uracils incorporated into viral DNA are sensed as a “DNA damage signal” and are excised by the uracil-DNA glycosylases (UDGs). Resulting viral DNA molecules harbor gaps and breaks that induce the DNA damage response in infected cells. Interestingly, the activation of DNA damage response is associated with an upregulation of NKG2D ligands on the surface of infected cells (Figure 2). Therefore, infected cells are recognized and killed by NK cells. The A3G knock-down by shRNA significantly improved survival of infected primary T cells co-incubated with autologous NK cells demonstrating that A3G sensitizes infected cells to NK lysis and promotes this innate immune pathway. It is important to note that Vif-mediated degradation of A3G is a protective mechanism developed by HIV to escape from this A3G antiviral activity. Therefore, the editing of HIV genome by A3G can be considered as “a danger signal” leading to innate immune responses. Moreover, several studies highlight that A3G also promote adaptive immunity. For example, A3G-edited defective viruses, which are not infectious, are able to activate HIV-1-specific CD8+ cytotoxic T lymphocytes (CTLs) in vitro [264]. In this model, the editing activity of A3G favors the generation of HIV-1 antigenic peptides by infected cells enhancing activation of HIV-1-specific CTL responses (Figure 2). A3G-mediated editing of viral genomes has been described to improve the capacity of infected DCs, which are professional antigen-presenting cells, to prime antiviral CTL responses [265]. However, since the editing activity of A3G is reduced in vivo mainly due to Vif counteraction, it only leads to low-level non-lethal mutations of the viral genome. In this context, the impact of A3G on adaptive immunity could be different. Interestingly, a study made in a cohort of HIV-infected individuals showed that A3G-mediated mutations are more abundant in sequences encoding immunodominant CD8+ T cells epitopes and make these epitopes less immunogenic [266,267]. This illustrates that HIV could have evolved to hijack the editing activity of A3G in order to decrease CTL recognition and escape from immune responses.

#### 4.1.4. BST-2

As mentioned above, two isoforms of BST-2 co-exist in cells. Interestingly, the longer isoform, but not the shorter, is an activator of the NF-κB pathway. Activation of NF-κB requires a tyrosine-based motif found within the cytoplasmic tail of the longer isoform [200]. BST-2 interacts with TAK1 and TAB1 to induce the canonical pathway of NF-κB activation [268]. In fact, signaling requires both the cytoplasmic tail of BST-2 and its extracellular domain involved in the retention of virions [269] showing that BST-2 activates NF-κB in response to the recognition of tethered virions (Figure 2). Thus, in addition to its role in viral restriction, BST-2 acts as a sensor of HIV-1 infection leading to pro-inflammatory responses. However, HIV-1 can escape from this sensing thanks to Vpu which inhibits BST-2 restriction as well as BST-2-induced NF-κB [268,270].

Since BST-2 can lead to the endocytosis of virions [195,199], it has been proposed that BST-2-mediated virus internalization could feed endosomal TLRs with viral PAMPs. For example, endocytosis of HIV-1 in pDCs allows the delivery of viral nucleic acids onto TLR7 leading to its activation and innate immune responses [23]. Thus, BST-2-mediated endocytosis of virions could be another mechanism to detect the presence of viruses. In line with this hypothesis, BST-2-mediated endocytosis of Friend retrovirus (FV) in mice models, activates DCs and leads to stronger NK cells and CD8+ T cell responses [271,272].

Interestingly, BST-2 has also been described to sensitize HIV-1 infected cells to antibody-dependent cellular cytotoxicity (ADCC) [273,274]. Indeed, mutations in Vpu (impairing BST-2 counteraction) or upregulation of BST-2 by IFN-β or IL-27 lead to an increase of HIV-env expression at the surface of infected cells favoring ADCC. On the opposite, knockdown of BST-2 decreases the susceptibility of HIV-infected cells to ADCC. Thus, in addition to its role as a restriction factor and in innate immunity, BST-2 could be connected with adaptive immunity by making virus-infected cells sensitive to antibodies.

Recently, it has also been shown that the upregulation of BST-2 by IFN-I sensitizes HIV-1 infected cells to NK cells responses [275]. In fact, this action of BST-2 is indirect and relies on Vpu. Vpu is able to promote NK cell evasion by downmodulating NTB-A (NK, T, and B cell antigen) and PVR (polio virus receptor), two ligands of the NK cells receptors NTB-A and DNAM-1 (DNAX accessory molecule 1). When BST-2 is upregulated, the occupation of Vpu by BST-2 competes with the ability of Vpu to downregulate NTB-A and PVR leading to recognition of infected cells by NK cells [275].

#### 4.1.5. Zinc-Finger Antiviral Protein (ZAP)

ZAP and more precisely the shorter isoform ZAP-S has been described as a positive regulator of innate immune responses. ZAP-S can associate with the cytoplasmic PRR RIG-I to promote the oligomerization and ATPase activity of RIG-I, leading to activation of IRF3 and NF-κB transcription factors [230]. SiRNA against ZAP-S impair induction of IFN-α and IFN-β after viral infection. However, deficiency of ZAP in mouse embryonic fibroblasts (MEF) does not affect RIG-I dependent IFN-I production in response to Newcastle disease virus (NDV) and IAV, making the role of ZAP in regulating RIG-I unclear [276]. Therefore, additional studies would be helpful to better determine the involvement of ZAP in immunity.

#### 4.1.6. Guanylate-Binding Protein 5 (GBP5)

After its induction by IFN, GBP5 is able to stimulate innate immunity by activating the NLRP3 and AIM2 inflammasomes [277,278]. Shenoy and colleagues have demonstrated that GBP5 has the unique property to activate NLRP3 inflammasome responses to pathogenic bacteria without being a NLR protein [277]. GBP5 and GBP2 also activate the AIM2 inflammasome (that recognizes dsDNA in the cytosol) in response to bacterial infection [278]. Inflammasome activation has been observed in HIV infected cells in particular NLRP3 activation [47,49]. In this context, it would be interesting to analyze if GBP5 may contribute to its activation. In addition to inflammasome activation, GBP5 is also able to stimulate the NF-κB signaling pathway enhancing IFN expression and IFN-related effectors [279]. Thus, after activation by innate immunity, GBP5 may in return, amplify this immune response.

### 4.2. Restriction Factors Negatively Regulating Immunity

#### 4.2.1. SAMHD1

The role of SAMHD1 as a regulator of the dNTPs pool was identified in patients affected by Aicardi–Goutières syndrome, an encephalopathic autoimmune disease, characterized by overexpression of IFN-I and IFN-associated genes [146]. In these patients, homozygous mutation of SAMHD1 gene leads to the accumulation of intracellular dNTPs that are sensed by PRRs, triggering IFN-I production [280,281]. This highlights that SAMHD1 acts as a negative regulator of innate immunity. In the case of HIV-1 infection, it has been described that SAMHD1 impairs cell-to-cell transmission of HIV-1 in moDCs limiting innate immune responses [282]. Silencing of SAMHD1 leads to HIV-1 sensing and induction of ISGs such as MxA. Moreover, moDCs do not produce detectable levels of IFN-I when they are exposed to cell-free HIV-1, whereas the addition of Vpx (which leads to inhibition of SAMHD1) induces the sensing of HIV-1 and promotes IFN production [282]. Furthermore, SAMHD1 prevents virus replication and thus the generation of reverse transcription products in DCs which decreases their ability to sense HIV-1 infection and to trigger their activation [41]. Generation of cGAMP, in response to HIV-1 infection and their sensing by the adaptor protein STING to induce IFN-I, can only occur in primary human DCs or macrophages under permissive conditions when SAMHD1 is inhibited by SIV-derived Vpx [40]. Recently, Chen and colleagues identified molecular pathways used by SAMHD1 to negatively regulate innate immune responses [283]. To prevent innate immunity to viral infections and inflammatory stimuli, SAMHD1 inhibits NF-κB activation by reducing phosphorylation of the NF-κB inhibitory protein IκBα (Figure 2). In addition, SAMHD1 also suppresses the IFN-I induction pathway by reducing IKKε-mediated IRF7 phosphorylation [283].

Thus, as described by others, HIV-1 might evade innate sensing through SAMHD1 [41]. DCs are cells that are hard to be productively infected by HIV-1 due to the restriction exerted by SAMHD1 [4]. However, DCs can internalize HIV-1 in immunoamphisomes where TLRs can be activated [30]. In this context, SAMHD1 negative regulation of innate immunity could be used as a mechanism to bypass host innate immunity. In the same way, HIV-1 could escape from adaptive immune responses since SAMHD1 has been described to limit HIV-1 antigen presentation by moDC [284].

In conclusion, SAMHD1 is a double-edged sword during HIV-1 infection since on one hand, it contributes to HIV-1 restriction, but on the other hand it might compete with the activation of innate and adaptive immune responses.

#### 4.2.2. BST-2

Despite many roles in promoting innate and adaptive immunity, BST-2 also has the ability to negatively regulate immune responses. BST-2 has been identified as a ligand for the immunoglobulin-like transcript 7 (ILT7), an inhibitory receptor expressed on human pDCs that regulates the production of IFN-I by these cells [285]. BST-2 binding to ILT7 inhibits IFN-I production following TLR7 and TLR9 activation (Figure 2). In the context of HIV-1 infection, Vpu suppresses TLR7-mediated IFN-I production by pDCs through a mechanism that relies on the interaction of BST-2 on HIV-producing cells with ILT7 on pDCs [286]. Indeed, even if Vpu down-regulates BST-2 surface expression, it also leads to its relocalization outside viral assembly sites where BST-2 can bind to ILT7 expressed on pDCs upon cell-to-cell contact. By this BST-2 mediated mechanism, Vpu limits the production of IFN-I by pDCs. Recent studies have shown that BST-2 can also directly inhibit RLR-mediated type I IFN signaling by targeting the mitochondrial protein MAVS for autophagic degradation [287]. In this context, BST-2 interacts with MAVS expressed on mitochondria and recruits the ubiquitin ligase MARCH8 (membrane associated ring-CH-type finger 8). Then, MARCH8 catalyzes the K27-linked ubiquitination of MAVS. This type of ubiquitination is a signal for selective degradation by autophagy since the autophagic receptor NDP52 binds to ubiquitinated MAVS proteins and targets them into autophagosomes. Importantly, co-localization between MAVS and BST-2 is enhanced upon TLR activation with poly (I:C) showing that the negative regulation of BST-2 is induced after the triggering of innate immune responses to regulate them in a negative feedback loop [287].

In conclusion, BST-2 is much more than a viral restriction factor since it plays a dual role in antiviral immune responses by promoting them on one hand, but also by acting as a negative regulator of IFN-I production on the other hand. This latter property could be used by HIV to escape from antiviral innate responses.

#### 4.2.3. Cholesterol-25-Hydroxylase

CH25H has been described as being able to negatively regulate immunity in return for its induction by IFN.

First, in naïve B cells, treatment with 25-HC suppresses B cell proliferation in response to IL-2- stimulation and blocks class switch recombination, leading to markedly decreased IgA production [210]. Secondly, in monkey-derived macrophages and PBMC cells, SIV infection induces CH25H expression via IFN which inhibits SIV replication but, in return, 25-HC produced by CH25H restrict mitogen-stimulated proliferation of cells and inhibit inflammation [288].

Interestingly, 25-HC increases the frequency of IFN-γ secreting cells activated in response to the SIV vaccine but selectively decreases the frequency of CD4+ T cells secreting IL-2, TNF-α and dual IL- 2/TNF-α cytokines, suggesting a negative regulation of the proinflammatory responses mediated by Th1 cells [288]. Taken together, these data suggest that, in addition to its antiviral role, CH25H regulates immunity in a negative feedback loop thanks to the production of 25-HC.

## 5. Conclusions

Host restriction factors play a crucial role in impairing HIV infection since these antiviral factors constitute a cellular intrinsic defense by blocking different steps of the HIV cycle. Viral restriction is so powerful that HIV needs to counteract these cellular factors for the establishment of a productive infection. For instance, viral accessory proteins Vif and Vpu inhibit A3G and BST-2 antiviral functions, respectively. HIV-1 can also avoid restriction factors through mutations as is the case for the HIV-1 capsid that evolved to escape from TRIM5α recognition (Table 1).

As we described in this review, most of restriction factors inducible by IFN-I and produced in response to the sensing of infection by PRRs, can be defined as classical ISGs. In addition, other mediators of innate immunity such as the type II IFNγ or interleukins can upregulate restriction factors expression (Table 1). Thus, restriction factors are not just intrinsic defense factors but are integral players of the innate immune response. Furthermore, some restriction factors, either through their antiviral functions or by additional mechanisms, modulate both innate and adaptive immune responses in positive or negative feedback loops (Table 1). As a consequence, the counteraction of restriction factors by HIV can impact immunity. Taken together, this review highlights the complex interplay between restriction factors and immunity that should be taken into account for further studies on HIV.

## Figures and Tables

**Figure 1 cells-08-00922-f001:**
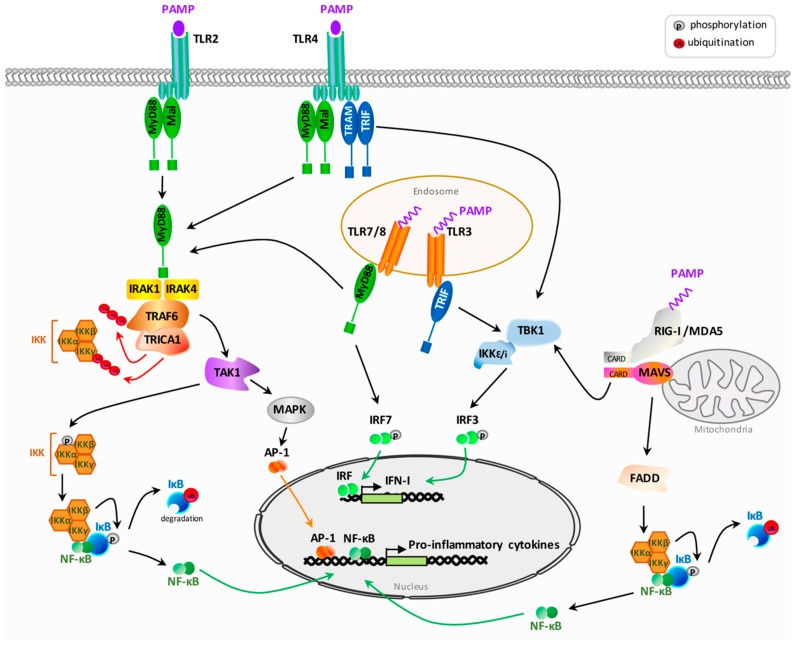
Simplified overview of several pattern recognition receptors (PRRs) signaling after pathogen-associated-molecular patterns (PAMPs) recognition. Signal transduction cascades leading to the production of type I interferon (IFN-I) and pro-inflammatory cytokines after Toll-like-receptors (TLRs) and retinoic acid-inducible gene I (RIG-I)-like receptors (RLRs) activation are described. See the text for more details. MyD88: myeloid differentiation factor 88, MAL: MyD88-adaptor-like, TRIF: TIR domain-containing adaptor-inducing IFN-β, TRAM: TRIF-related adaptor molecule, IRAK: interleukin-1 receptor-associated kinase, TRAF6: TNF receptor associated factor 6, TAK1: transforming growth factor beta-activated kinase 1, MAPK: mitogen-activated protein kinase, IKK: IκB kinase, NF-κB: nuclear factor-kappa B, IRF: IFN-regulatory factor, TBK1: tank-binding kinase 1, MAVS: mitochondrial antiviral-signaling, FADD: Fas-associated protein with death domain.

**Figure 2 cells-08-00922-f002:**
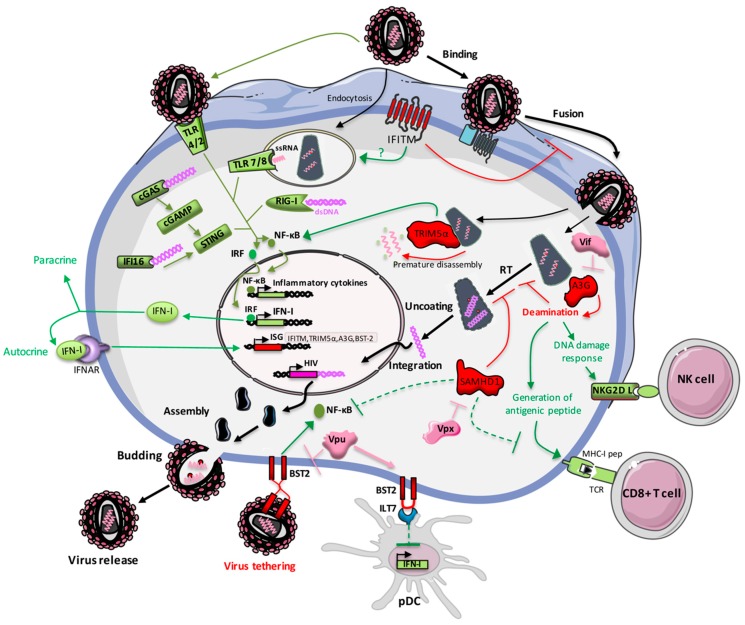
Schematic representation of HIV sensing by PRRs leading to the expression of antiviral restriction factors and shaping of immunity by restriction factors. Detection of HIV PAMPs by PRRs leads to activation of the transcription factors IRF and NF-κB driving the expression of IFN-I and inflammatory cytokines, respectively. IFN-I binds to IFNAR (IFN-α/β receptor chain) and activates transcription of interferon-stimulated genes (ISGs) including the restriction factors interferon-inducible transmembrane (IFITM) proteins, tripartite motif (TRIM)5α, apolipoprotein B mRNA-editing enzyme catalytic subunit-like 3G (APOBEC3G/A3G) and bone marrow stromal antigen 2 (BST-2). Antiviral functions of restriction factors are indicated by red arrows. Feedback of restriction factors on immunity are depicted by green arrows (induction: solid lines, inhibition: dotted lines). Counteractions by viral proteins are depicted in pink. Only restriction factors that are both induced by immunity and have an effect on immunity are illustrated. pDC: plasmacytoid dendritic cells, NK: natural killer cells, RT: reverse transcription, TLR: Toll-like receptors, RIG-I: retinoic acid-inducible gene I, cGAS: cyclic guanosine monophosphate (GMP)-adenosine monophosphate (AMP) synthase, cGAMP: cyclic GMP-AMP, STING: stimulator of interferon genes, IFI16: interferon gamma inducible protein 16, IRF: IFN-regulatory factor, ssRNA: single-stranded RNA, dsDNA: double-stranded DNA, TCR: T cell receptor, MHC-I pep: major histocompatibility complex I associated with a peptide.

**Table 1 cells-08-00922-t001:** Overview of the interplay between restriction factors and immunity.

Restriction Factor	Induction by Innate Immunity	Antiviral Function	Feedback on Immunity	HIV Counteraction
**IFITM1/2/3**	IFN-α IFN-γ	- Inhibit viral entry by reducing membrane fluidity - Negative imprinting of virions	- Involved in Th1/Th2 polarization of CD4+ T cells - Targeting of virus in endosomes for TLRs sensing?	
**TRIM5α**	IFN-I	- Premature uncoating - Targets viral capsid for proteasomal degradation	- Activation of NF-κB after sensing the viral capsid	Viral capsids with reduced affinity for TRIM5α
**APOBEC3G**	IFN-α IFN-γ IL-2, IL17, IL15	- Deamination of cytidines to uracils during RT creating hypermutated proviral DNA	- Induction of NKG2D ligand expression leading to the recognition of infected cells by NK cells - Generation of antigenic peptides presented on MHC-I allowing the recognition of infected cells by CD8 T cells	HIV-1 Vif
**SAMHD1**	TLR3, RIG-I and MDA5 activation (in HeLa, HEK293 and MARC-145 cells) IFN-α (liver cells) IL-12, IL18	- Inhibits RT by decreasing the cellular pool of dNTPs - Degradation of HIV genomic RNA	- Inhibition of NF-κB - Decreases antigen presentation	HIV-2 Vpx
**Mx2**	IFN-α, IFN-β IFN-γ	- Inhibits HIV nuclear import - Impairs uncoating?	Not described	
**BST-2/Tetherin**	IFN-α IFN-γ IL27	- HIV budding (viral entrapment) - Internalization and degradation of virions by the endosomal pathway	- Activation of NF-κB - Delivery of PAMPs on endosomal TLRs? - Sensitizes infected cells to ADCC - Sensitizes infected cells to NK cells - Binds to ILT7 on pDCs and inhibits IFN-I production - Inhibits RIG-I signaling (leads to the autophagic degradation of MAVS)	HIV-1 Vpu
**CH25**	IFN-α, IFN-β IFN-γ TLR4/TLR3 activation	- Inhibits viral entry by affecting membrane fluidity	- Inhibits inflammation - Blocks Ig class switch recombination in B cells	
**ZAP**	IFN-α	- Degradation of viral mRNA - Inhibition of translation	- ZAP-S promotes RIG-I activity	HIV-Rev via Matrin3
**SLFN11**	IFN-I	- Inhibits HIV translation by preventing the change of tRNA pool composition	Not described	
**ISG15**	IFN-α, IFN-β NF-κB	- Inhibits HIV-1 release (ISGylation of Gag)	Not described	
**GBP5**	IFN-I	- Decreases viral progeny infectivity by impairing incorporation of gp120 in budding viruses	- Activates NLRP3 and AIM2 inflammasomes - Stimulation of NF-κB

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
