# Peer review of "Interplay between Intrinsic and Innate Immunity during HIV Infection"

_cells, 2019, doi:10.3390/cells8080922_

Round 1
Reviewer 1 Report
This manuscript is a comprehensive review of intrinsic and innate immunity of HIV infection. It is a timely review given the numerous advances in the field, and it will be quite useful. The manuscript is well organized and well presented. The literature cited is exhaustive. A number of suggestions for improving the manuscript are listed below. There are several grammatical and stylistic errors of English throughout the manuscript, but presumably a careful proofreading by the Authors and copy editor will correct them.
Line 51. “… since HIV-1 proteins are able to counteract most of restriction factors ..”
A more accurate statement would be “many” rather than “most.”
Figure 1. The Authors should consider incorporating the RIG-I/MDA5 pathway in the figure. Figure 2. What does the “LT CD8” label mean? Perhaps this is a standard French abbreviation for CD8 T cells, but it will be confusing to many readers. ZIKA should be Zika (it is not an abbreviation such as HCV or HIV). Line 280: Trim5α should be TRIM (?)
Author Response
"Please see the attachment."

Reviewer 2 Report
This is a very comprehensive and extensive review providing a detailed description of different restriction factors. The authors make an effort to show that some restriction factors may have contrasting effects on the development of the immune response.
In general, the different restriction factors and ISGs are very well described, referring to the most important recent literature. In this sense, I think the authors did a very good job.
We have two suggestions to improve the overall clarity of the review
1) As the title and the abstract imply, the manuscript is supposed to describe the interplay between intrinsic and innate immunity. Indeed, the manuscript shows that retroviral restriction factors are also inducible by IFN and trigger a response contributing to the development of downstream immunity. The doubt therefore arise. What is the difference between intrinsic and innate immunity? Perhaps it would be useful to spend a few sentences defining these concepts since a definition is never given in the manuscript.
2) While we understand the effort of the authors to compartmentalize the manuscript in chapters describing different effects on immunity, having to read distinct paragraphs about BST2, SAMHDI, APOBEC3, IFITM, TRIM5 etc… two or three times in different points of the manuscript appears redundant and somewhat confusing. Would it be possible, instead, to unify by restriction factor and explain the opposing effects of each restriction factor in the same paragraph?
Author Response
"Please see the attachment."

Reviewer 3 Report
The authors have provided a detailed comprehensive summary of the role of restriction factors in human immunodeficiency virus infection with considerable information on the role of these factors in other viral infections presented alongside as literature review or to illustrate their potential mechanism(s) of action. This itself is high quality and of interest to readers and the authors also then comprehensively reviewed the links between restriction factors, innate and in some cases adaptive immunity in a follow-up section. The linkage to interferons, especially type I, was stressed throughout and the complexity of interrelationships between 3 branches of immunity was well represented. Overall this is an excellent comprehensive review and I have only a few comments/suggestions for improvement.
In some cases, the review might actually be too comprehensive in that results from a single study are cited as evidence for the activity that may or may not be relevant or meaningful. The material is presented objectively, but it may help at times to point out the limitations in the experimental data and suggest that things seen in particular experiments aren't necessarily representative of an antiviral role in vivo.
The sentence on lines 140-143 is difficult to follow.
Lines 193-199 are also difficult to follow. Is the HIV capsid in the nucleus of infected cells? Why HIV-1 and not HIV-2?
In the section describing links between APOBEC3G activities and innate/adaptive immunity, the authors should discuss the work of Larijani and others indicating an enrichment of target signs within sequences composing CD8+ T cell epitopes and A3G mutations leading to immune escape as an opposing view as to how HIV has evolved to exploit APOBEC activity.
There are very minor instances of improper English usage distributed throughout the manuscript (s out of place, tense, some word meanings). Although these do not affect the reader's comprehension of the material, an English language editor could very quickly eliminate the errors and improve the manuscript.
Author Response
"Please see the attachment.
